# Family history of Alzheimer's disease alters cognition and is modified by medical and genetic factors

Joshua S Talboom[1,2], Asta Håberg[3], Matthew D De Both[1,2], Marcus A Naymik[1,2], Isabelle Schrauwen[1,2], Candace R Lewis[1,2], Stacy F Bertinelli[1], Callie Hammersland[1], Mason A Fritz[1], Amanda J Myers[4], Meredith Hay[2,5], Carol A Barnes[2,5], Elizabeth Glisky[2,5], Lee Ryan[2,5], Matthew J Huentelman[1,2]*

[1]The Translational Genomics Research Institute, Phoenix, United States; [2]Arizona Alzheimer's Consortium, Phoenix, United States; [3]Norwegian University of Science and Technology, Trondheim, Norway; [4]University of Miami, Miami, United States; [5]University of Arizona, Tucson, United States

**Abstract** In humans, a first-degree family history of dementia (FH) is a well-documented risk factor for Alzheimer's disease (AD); however, the influence of FH on cognition across the lifespan is poorly understood. To address this issue, we developed an internet-based paired-associates learning (PAL) task and tested 59,571 participants between the ages of 18–85. FH was associated with lower PAL performance in both sexes under 65 years old. Modifiers of this effect of FH on PAL performance included age, sex, education, and diabetes. The Apolipoprotein E ε4 allele was also associated with lower PAL scores in FH positive individuals. Here we show, FH is associated with reduced PAL performance four decades before the typical onset of AD; additionally, several heritable and non-heritable modifiers of this effect were identified.
DOI: https://doi.org/10.7554/eLife.46179.001

*For correspondence:
mhuentelman@tgen.org

Competing interests: The authors declare that no competing interests exist.

## Introduction

Alzheimer's disease (AD), the leading cause of dementia, is a progressive neurodegenerative disorder that typically first presents clinically as deficits in cognition. It is estimated that over five million people in the United States are currently living with AD, and by 2050, that number is expected to climb to 13.8 million (*Hebert et al., 2013*). The two major risk factors for the more common late-onset form of AD are increasing age and a first-degree family history of dementia (FH) (*Braak et al., 2011*; *Green et al., 2002*). FH is known to encompass both heritable and nonheritable risk factors for AD (*Chang et al., 2012*; *Yi et al., 2018*). FH has been associated with changes in multiple cognitive domains previously in children (13 years old) and young adults (35 years old); however, fewer cognitive domain changes were reported in middle-age (53 years old) and older (65–78 years old) FH adults (*Aschenbrenner et al., 2016*; *Bloss et al., 2008*; *Honea et al., 2009*; *La Rue et al., 2008*; *Miller et al., 2005*; *Parra et al., 2015*; *Zeng et al., 2013*). These effects were unrelated to the apolipoprotein (APOE) ε4 allele. Despite these FH findings, to the authors' knowledge, there has not been a well-powered study of the effects of FH on memory across the lifespan. Furthermore, it is unknown if FH status interacts with demographics, common health and APOE genotype. Identification of factors potentiating or ameliorating an effect of FH is particularly crucial as one approach to identifying new avenues of risk reduction for AD. Indeed, risk reduction for AD is now more critical than ever due to the continued lack of a cure or effective disease-slowing treatments for AD (*Barnes and Yaffe, 2011*; *Sperling et al., 2011*; *Takeda and Morishita, 2018*). This fact underscores

risk reduction as one of the most practical means to attenuate the enormous burden AD places on society (*Baumgart et al., 2015*).

To overcome challenges associated with underpowered studies, we created MindCrowd (www.mindcrowd.org) as an accessible and easy-to-use web-based cognitive and demographic assessment; specifically, to elucidate the effects of FH on cognition in healthy participants. Participants were recruited across the entire adult lifespan to understand better how FH may alter cognition at different ages. The cognitive assessment consisted of an online, verbal paired-associates learning (PAL) task. PAL was chosen because it is a medial temporal lobe (MTL)-dependent learning and memory task affected early in the onset of AD (*Pike et al., 2008*). In addition, PAL is sensitive to changes in performance associated with healthy aging (*Pike et al., 2013*). Data on multiple demographic, medical, health, and lifestyle factors were also obtained (see *Supplementary file 2*). FH status was determined by self-report of having (FH+), or not having (FH-) based on the question, 'Have you, a sibling, or one of your parents been diagnosed with Alzheimer's disease?' Further, a subset of FH participants was solicited to send us dried blood spots so that we could determine their apolipoprotein (APOE) genotype. APOE is a well-known genetic risk factor for AD. The risk of developing AD is higher by ~6% in ε4 allele carriers, while total genetic heritability risk for AD is 33% based on single nucleotide polymorphism data (*Ridge et al., 2013*), and ~60–75% in twin studies (*Reynolds and Finkel, 2015*). In addition, a recent study found a synergistic effect of FH and APOE ε4 on higher levels of amyloid-beta deposition measured via positron emission tomography (PET) imaging (*Yi et al., 2018*). As a result of these data and numerous other reports, we hypothesized that the APOE ε4 allele would relate to lower PAL scores in FH participants.

## Results

As of August 15, 2018, MindCrowd, has recruited 59,571 qualified participants (see Data Quality Control in Materials and methods) from around the world (*Figure 1A*). The sample was 62.46% female and 37.54% male (*Figure 1B*). An overrepresentation of women has been previously described in studies drawn from the general population (*Krokstad et al., 2013*) as well as for AD (*Roberts et al., 2004*). The breakdown of race was American Indian or Alaska Native = 0.62%, Asian = 5.13%, Black/African American = 1.75%, Native Hawaiian/Pacific Islander = 0.39%, Mixed = 0.0009%, and White = 92.03%. In terms of years of education, we collapsed across education milestones for visualization purposes. Here we found that 10.78% reported ≤12 y, 29.64% reported ≤14 y, 35.32% ≤ 16 y, and 24.25% reported ≥20 y (*Figure 1C*). Across the entire sample, a FH of AD is present in 22.76% with the overall percentage swelling with age (*Figure 1D*).

In women and men 18–85 years old, our general linear model (GLM, see *Figure 1—figure supplement 1* for regression diagnostic plots and *Supplementary file 1* for a table all coefficient $n$s) revealed a significant Age coefficient ($B_{Age} = -0.20$ word pairs, $p_{Age}$ <2e-16). Age was associated with a lowered PAL performance of two word-pairs per decade of life (*Figure 2*). Sex was also a significant predictor of PAL scores ($B_{Sex} = -1.82$, $p_{Sex}$ <2e-16). Women were associated with nearly a two word-pair higher PAL score compared to men (*Figure 2*). Propensity score matching (PSM, see Statistical Methods in Materials and methods) revealed that the associated disparity between women and men's PAL scores markedly grew around the 5[th] decade of life. PSM revealed an estimated effect size (i.e., average treatment effect among treated, ATT) of being a woman grew from the 5[th] to the 6[th] decade of life (50 s: $2.68_{ATT} \pm 0.35$ $_S$D-word pairs, 60 s: $3.72_{ATT} \pm 0.32$ $_S$D-word pairs). Educational Attainment was another significant predictor of PAL scores ($B_{Education} = 0.31$, $p_{Education}$ <2e-16, *Figure 3*). For both women and men, each milestone of Educational Attainment was associated with around a third of a word pair higher PAL score. In women and men, PSM revealed an approximate effect size of one-word pair higher performance across each level of Educational Attainment (e.g., Male 12y vs. 14y [$_{ATT}0.72 \pm$ $_{SD}0.28$], Female 12y vs. 14y= [$_{ATT}1.06 \pm$ $_{SD}0.24$]). However, the magnitude of this effect was not consistent for men and women across levels of Educational Attainment. Indeed, PSM estimated that women have a higher PAL score related to education than men, at all except the highest level of Educational Attainment (16y vs. 20y: Women=[$_{ATT}0.97 \pm$ $_{SD}0.12$], Men=[$_{ATT}1.29 \pm$ $_{SD}0.16$]).

Notably, there was a significant main effect of FH ($B_{FH} = -2.39$, $p_{FH} = 3.47e-06$, *Figure 4A*.). This coefficient indicated that for women and men, having FH was associated with nearly a two-and-a-half-word pair lower PAL scores when compared to FH- participants. Further our model revealed a

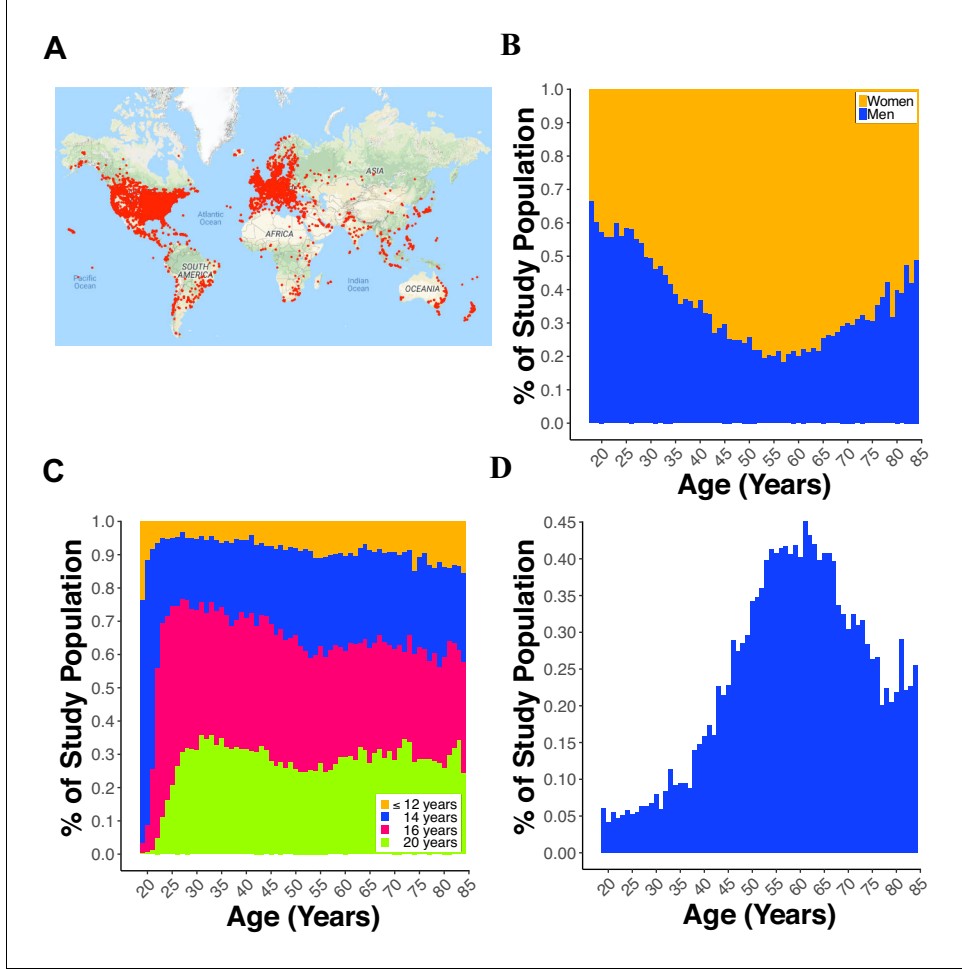

**Figure 1.** Demographics of participants. (A) World map displaying a red dot (i.e., one dot = one IP address) at the location of a participant completing the paired-associates learning (PAL) task. (B) Line plot showing the percent of males and females from 18 to 85 years old. (C) Line plot displaying the percent of participants with 12, 14, 16, or 20 years of education for each year of age from 18 to 85 years old. (D) Line plot showing the percent of participants reporting a first-degree family history of Alzheimer's disease (FH) for each year of age from 18 to 85 years old.

DOI: https://doi.org/10.7554/eLife.46179.002

The following figure supplements are available for figure 1:

**Figure supplement 1.** Regression diagnostic plots of the general linear model (GLM) including all participants (N=59,571).

DOI: https://doi.org/10.7554/eLife.46179.003

**Figure supplement 2.** Simulated additional self-report error and the impact on the significance of the FH effect in MindCrowd.

DOI: https://doi.org/10.7554/eLife.46179.004

significant Age x Diabetes ($B_{Age*Diabetes}$ = 0.03, $p_{Age*Diabetes}$ = 0.03) and Age x FH ($B_{Age*FH}$ = 0.02, $p_{Age*FH}$ <0.01, *Figure 4A*) interaction. The significant interactions between Age and Diabetes as well as Age and FH indicate a greater association between Diabetes and FH with lower PAL scores at younger ages as compared to older ages. Indeed, the linear trend lines for diabetes cross at 50 years of age (data not shown) and at age 65 for FH (*Figure 4A*). Due to the significant Age x FH interaction, our analyses evaluating interactions with FH included only participants $\leq$ 65 years old. A significant Sex x FH interaction ($B_{Sex*FH}$ = −0.79, $p_{Sex*FH}$ = 7.97e-06, *Figure 4B*) was found. Follow-up analyses of the estimated marginal mean (EMM, see Statistical Methods in Materials and methods) revealed that FH +women and men had lower PAL scores compared to FH- women and men

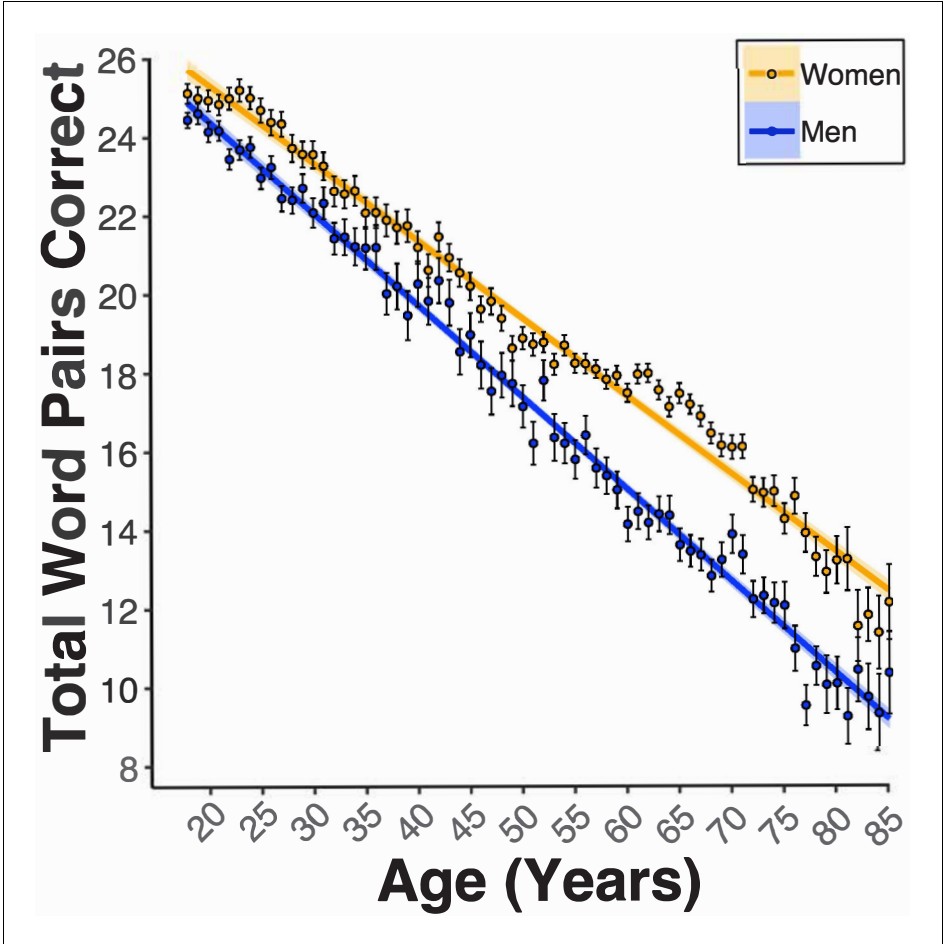

**Figure 2.** Females demonstrate enhanced paired-associates learning (PAL) performance across the aging spectrum, and this is further enlarged beginning in the 5th decade of life. Linear regression fit (line fill ±95% confidence interval [CI], error bars ± standard error of the mean [SEM]) of the PAL total number of correct from 18 to 85 years old. Lines were split by Sex. Women performed better than men with an amplified disparity from 50 to 70 years old ($B_{Sex}$ = −1.82, $p_{Sex}$ <2e-16, women n = 40572, and men n = 24381).

DOI: https://doi.org/10.7554/eLife.46179.005

(women: EMM = 1.01, $t(53763)$ = 4.18, p<0.01; men: EMM = 1.58, $t(53763)$ = 6.03, p<0.001); however, the magnitude of this effect was different between women and men. Indeed, PSM revealed that the estimated word pair reduction in PAL scores associated with FH status was larger in men (e.g., 20 s: $_{ATT}$1.93 ± $_{SD}$0.60) as compared to women (e.g., 20 s:$_{ATT}$0.60 ± $_{SD}$0.67), except for the 6th decade of life (60 s: Women[$_{ATT}$-0.50 ± $_{SD}$0.18], Men[$_{ATT}$0.09 ± $_{SD}$−0.36], *Figure 4* Inset).

There was a significant Educational Attainment x FH interaction ($B_{Education*FH}$ = 0.08, $p_{Education*FH}$ <0.01). Follow-up analyses of the EMMs showed that FH was associated with lower PAL score at each milestone of educational attainment ($ts(53758)$ > 3.45, ps <0.01); the magnitude of the associated decline in PAL scores attributed to FH was greater at lower milestones of educational attainment as compared to higher milestones (e.g., 12y [High school diploma]: FH- EMM = 16.4, FH +EMM = 14.8, difference = 1.6; 20y [Post graduate degree]: FH- EMM = 20.8, FH +EMM = 19.6, difference = 1.2). In FH +or FH- women and men 18–65 years old, our model revealed a significant FH x Diabetes ($B_{FH*Diabetes}$ = −0.71, $p_{FH*Diabetes}$ = 0.04, *Figure 5*) interaction. Follow-up analyses of the EMM revealed that participants with FH and diabetes were associated with lower PAL scores compared to participants with FH but no diabetes (FH +Diabetes + EMM = 16.9, FH +Diabetes- EMM = 18.1, $t(53763)$ = 2.73, p=0.03). Lastly, two models which included either: 1) the Number of APOE ε4 alleles (i.e., 0, 1, or two copies) or 2) APOE Genotype (e.g., ε2-ε2, ε2-ε3, ε4-ε4, etc.) were

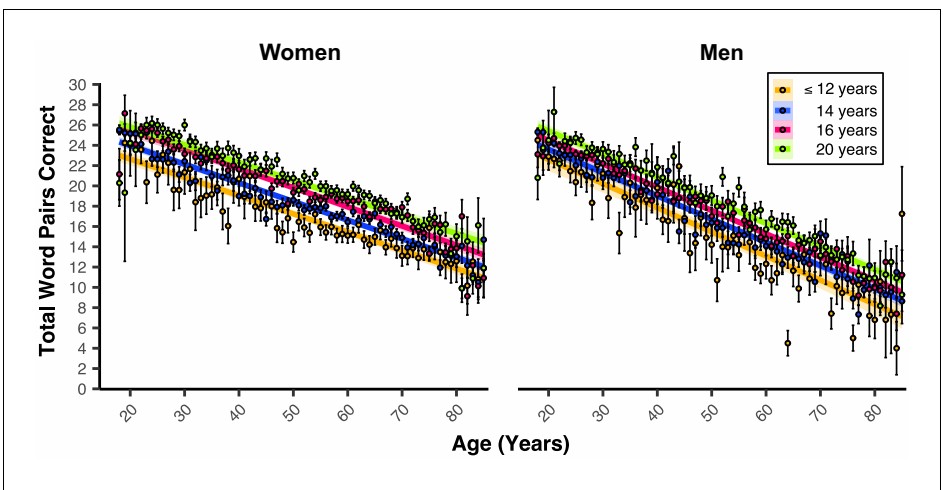

**Figure 3.** Educational attainment is associated with PAL performance in a milestone-related dose response that is different between the sexes. Linear regression fits (line fill ±95% CI, error bars ± SEM) of the PAL total number of correct from 18 to 85 years old. Lines were split by Educational Attainment level, and the figure was faceted by Sex. For women and men, there were heightened PAL scores per level of Educational Attainment ($B_{Education}$ = 0.31, $p_{Education}$ <2e-16, 6 years $n$ = 282, 8 years $n$ = 181, 10 years $n$ = 1177, 12 years $n$ = 5367, 14 years $n$ = 19256, 16 years $n$ = 22942, and 20 years $n$ = 15752).
DOI: https://doi.org/10.7554/eLife.46179.006

evaluated. There was a significant main effect for the Number of APOE ε4 Alleles variable ($B_{ε4Allele}$ = −1.30, $p_{ε4Allele}$ = 0.02, **Figure 6**). These data indicated that in FH +participants, there was an association of one and a third word pair lower PAL score per each ε4 allele. Moreover, there was a significant main effect in the model comparing the ε4-ε4 to the ε2-ε3 genotype ($B_{ε4\text{-}ε4\ vs.ε2\text{-}ε3}$−5.32, $p_{ε4\text{-}ε4\ vs.ε2\text{-}ε3}$30.03). Similarly, FH +participants with the ε4-ε4 genotype were associated with a five-word pair lower PAL scores compared to FH +participants with the ε2-ε3 genotype.

## Discussion

Overall, this study finds that having a first-degree relative diagnosed with Alzheimer's disease (FH) is associated with lower verbal learning and memory performance (i.e., paired-associates learning; PAL) below the age of 65. Notably, FH men showed a greater reduction in cognition compared to FH women. This effect is not surprising since women, regardless of FH, perform better on PAL compared to men (*Kaushanskaya et al., 2011*), an effect replicated in this study. To that end, we demonstrate the sex-effect for PAL extends across the entire adult lifespan; indeed, the first time to the authors' knowledge that this has been demonstrated: 1) in a large cohort, 2) using the same test, and 3) in a single study. It is interesting that our study found the associated disparity between women's and men's PAL scores enlarged around the 5th decade of life. The 5th decade of life is the approximate age when women undergo menopause in developed countries. Menopause-related changes to women's hormonal milieu, either endogenously or via hormone treatment or gynecological surgery have been found to alter cognition during this period (discussed in *Koebele and Bimonte-Nelson, 2016*). Future studies of this cohort will dissect which medical choices at menopause, and medical choices earlier in a woman's lifespan, may underlie better preservation of verbal memory in middle aged women as compared to men. Lastly, educational attainment was associated with milestone-dependent higher learning and memory performance. Regardless of FH status, education in women was associated with better PAL scores compared to men, at all except the highest reported level of educational attainment (i.e., 16 vs. 20 years of education). The effect of education on PAL scores is in line with observations in a clinical study demonstrating that higher levels of education are associated with delayed AD onset (*Vemuri et al., 2014*; *Wang et al., 2012*).

In terms of health, lifestyle, and genetic factors, this study found that diabetes modified the effect of FH on PAL. Specifically, in both women and men, diabetes and FH were associated with reduced

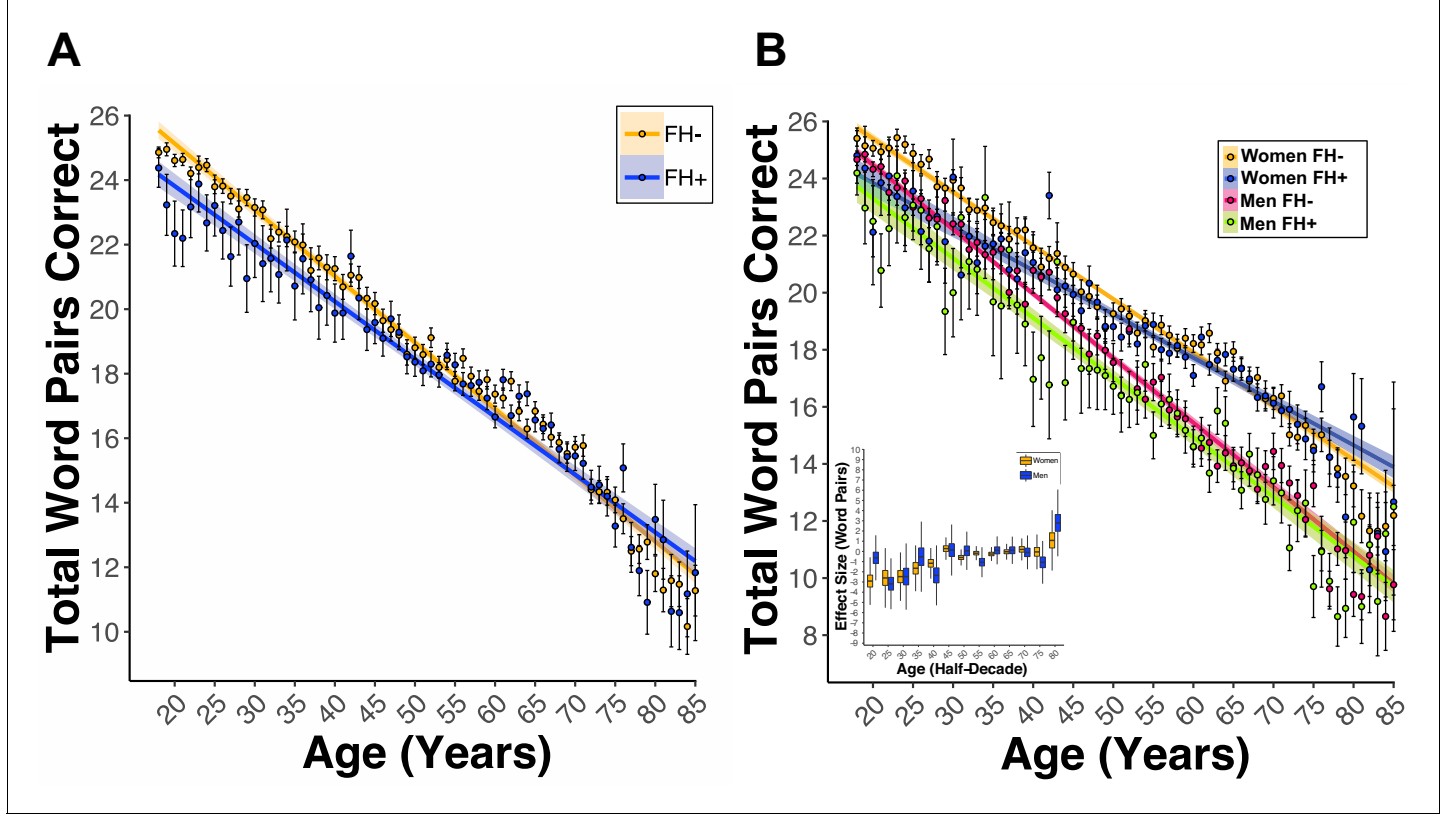

**Figure 4.** A first-degree relative FH of AD is associated with lower PAL performance at ages under 65. (**A**) Linear regression fits (line fill ±95% CI, error bars ± SEM) of the PAL total number of correct from 18 to 85 years old. FH led to lower PAL performance before 65 years of age ($B_{FH}$ = −2.39, $p_{FH}$ = 3.47e-06, FH +$n$ = 14739, and FH- $n$ = 50011). (**B**) Linear regression fits of the PAL total number of correct from 18 to 85 years old. Lines split by Sex and FH status. FH led to lower PAL performance, an effect that was exacerbated in men ($B_{Sex*FH}$ = −0.79, $p_{Sex*FH}$ = 7.97e-06, FH +Women $n$=11119, FH- Women $n$ = 29332, FH +Men $n$=3617, and FH- Men $n$ = 20678). The inset figure displays PSM box and whisker plots, split by sex, across each decade of life. The black bar through each box and whisker plot represents the median ATT for FH. Men had worse PAL scores when compared to women at each decade of life except for the sixth (60 s: Women[$_{ATT}$-0.50 ± $_{SD}$0.18], Men[$_{ATT}$0.09 ± $_{SD}$−0.36], women $n$ = 40572, and men $n$ = 24381).
DOI: https://doi.org/10.7554/eLife.46179.007

PAL performance. It is not surprising that diabetes exacerbates the effects of FH on cognition since diabetes has been linked to worse cognitive deficits in AD (*Takeda and Morishita, 2018*). Several factors may underlie this effect: 1) there are differences in the risk of dementia for type one diabetes and type two diabetes, 2) specific type two diabetes treatments may reduce age-related declines in neural metabolism (*Hamed, 2017*; *Kuo et al., 2018*), and 3) vascular damage may combine with other genetic and environmental factors. As for genetic factors that modify FH, we supported our hypothesis whereby the presence of an APOE ε4 allele was associated with lower PAL performance in a dose-dependent-like manner in FH individuals. These data suggest that APOE genotype is an important genetic factor that influences memory. Our findings are in line with results from a voxel-wise study in humans noting a synergistic effect of FH and the APOE ε4 allele to intensify amyloid-beta deposition and reduce glucose use in regions of the MTL and other AD-related brain regions (*Yi et al., 2018*).

At the systems level, these results suggest that the collection of heritable and non-heritable changes due to FH status alter the functioning of the MTL and associated structures (*Yi et al., 2018*). The fact that these effects were only observed in participants that were ≤65 years old could be due to age-related differences in the effect of FH or participation differences between younger and older FH test takers. The ≤65 years old effect is partially in line with earlier smaller and more age-specific studies of FH status and cognition demonstrating an effect of FH, but only in children and younger adults (*Bloss et al., 2008*; *Parra et al., 2015*; *Zeng et al., 2013*). Since the prevalence of AD rises after age 60, it is also possible that older FH participants consenting to our study are

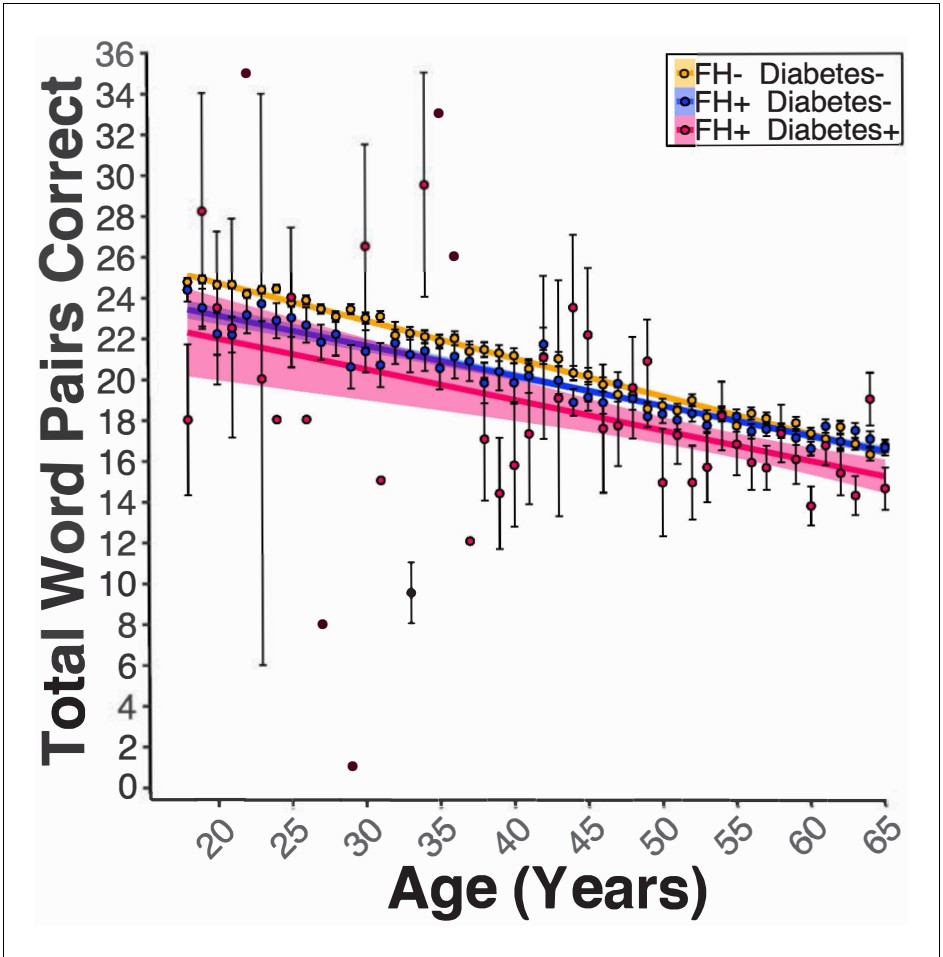

**Figure 5.** Diabetes modified the FH effect on PAL performance. Linear regression fits (line fill ±95% CI, error bars ± SEM) of the PAL total number of correct in FH- Diabetes+, FH +Diabetes-, and FH +Diabetes + participants from 18 to 65 years old. Regardless of sex, diabetes in FH +participants led to lower PAL scores ($B_{FH*Diabetes}$ = −0.71, $p_{FH*Diabetes}$ = 0.04, AD- DI- $n$ = 47970, AD- DI +$n$ = 2041, AD +DI n=13841, and AD +DI + n=898).

DOI: https://doi.org/10.7554/eLife.46179.008

those that have remained cognitively intact. Thus, participants that were experiencing noticeable age- or disease-related cognitive impairment may choose not to participate.

It is important to acknowledge some limitations of our work. FH risk is known to vary depending on the relationship of the diagnosed relative, and previous reports have demonstrated that first-degree FH results in higher risk for dementia compared to second- and third-degree FH (*Cannon-Albright et al., 2019*). In our study, we asked about the first-degree FH only; therefore, it is possible that individuals who have other FH risk from extended family members were included in our non-FH group. Additionally, it is possible that the form of AD, late-onset versus the rarer early-onset form, may encode different levels of FH risk. Future work is planned to investigate the FH effect in the study cohort, including an improved ability to separate late and early onset FH for each participant as well as to inquire about additional extended family member FH status.

It is likely that we did not measure all demographic, lifestyle, and health factors that are associated with differential PAL performance. One such example is socioeconomic status (SES). SES has been shown to have an association with brain structure and cognitive measures during development (reviewed in *Brito and Noble, 2014*) and work also suggests SES could play a role in AD risk (*Qian et al., 2014*; *Stępkowski et al., 2015*; reviewed in *Seifan et al., 2015*). Importantly, while we did not measure SES directly, we did assess factors commonly used to construct the SES composite

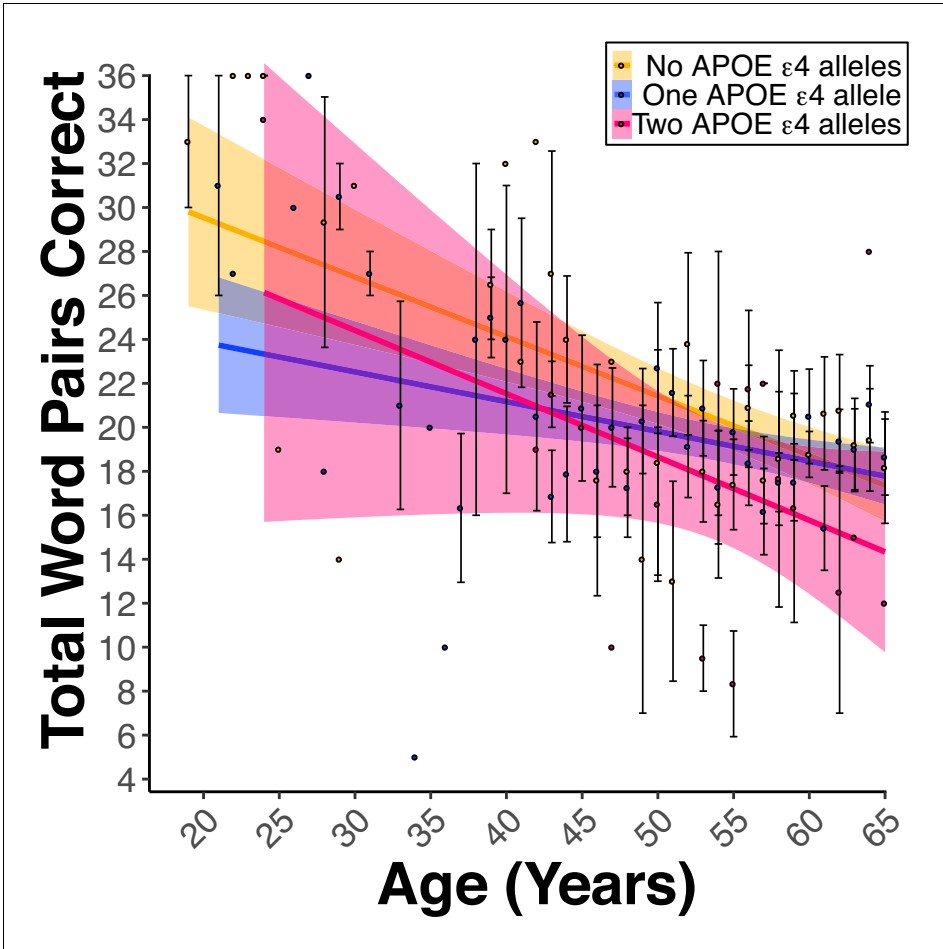

**Figure 6.** Apolipoprotein E (APOE) ε4 alleles negatively influence PAL performance in the presence of FH. (**A**) Linear regression fits (line fill ±95% CI, error bars ± SEM) of the PAL total number of correct from 18 to 65 years old. Lines were split by the Number of APOE ε4 Alleles. For women and men, there was a dose-dependent-like decrease in the PAL scores per each copy of the ε4 allele ($B_{ε4Allele}$ = −1.30, $p_{ε4Allele}$ = 0.03, ε2/ε2 $n$ = 2, ε2/ε3 $n$ = 31, ε2/ε4 $n$ = 46, ε3/ε3 $n$ = 174, ε3/ε4 $n$ = 382, and ε4/ε4 $n$ = 35).
DOI: https://doi.org/10.7554/eLife.46179.009

(e.g., Educational Attainment). In addition, due to the international recruitment of our study cohort, normalization of the SES construct is complicated due to differing definitions of the factors used to calculate SES across nations (*Rose et al., 2005*).

Due to the large, distributed, and electronic nature of our study cohort, we rely on self-report answers to demographic, lifestyle, and health questions. Current studies comparing self-report data given over the internet versus in-person collected data show anywhere from a 0.3–20% discrepancy for height and weight measurements (*Maukonen et al., 2018*; *Nikolaou et al., 2017*). To investigate the potential role that such error may play on our FH AD effect, we re-analyzed the FH effect after introducing additional error into the coding of the FH self-report response. Additional error was added by randomly re-assigning FH status to various percentages of the cohort (stepwise from 2–30% of individuals) and re-analyzing the effect of FH using our complete statistical model. This was performed a total of 10,000 times for each error percentage, and the resulting influences on the p-value were reported using boxplots. With 8% additional introduced error we are able to show statistically significant effect of FH on PAL in 100% of the 10,000 tested iterations while an additional 24% error in FH status would still result in a statistically significant effect of FH on PAL in over 50% of the iterations (see *Figure 1—figure supplement 2*). These results suggest that it is unlikely that FH self-report error is driving the significant effect of FH on PAL. Lastly, PAL was tested cross-

sectionally in the cohort; therefore, determinations about the influence of collected factors on trajectories of change in performance across time within an individual subject are not possible. Additional longitudinal-based studies will be necessary to identify this class of variables.

Collectively, this study supports recommendations underscoring the importance of living a healthy lifestyle, properly treating disease states such as diabetes, and building cognitive reserve through education (i.e., risk reduction) to attenuate age- and AD-risk-related cognitive declines. Further, our findings specifically highlight the positive effects of such interventions on FH-associated risk, opening the door to the development of more targeted risk reduction approaches to combat AD. In addition, this work underscores the utility of web-based cohort recruitment and study; thus, facilitating large sample sizes in a cost- and time-effective fashion. Statistical power concerns are common in many scientific studies, and underpowered study designs can lead to several potential issues associated with false positive and negative rates. It should be acknowledged that web-based studies are not without concerns, however, we propose that the advantage of considerably larger sample sizes and enriched cohort diversity in online research mostly diminishes the potential disadvantages.

# Materials and methods

**Key resources table**

| Reagent type (species) or resource | Designation | Source or reference | Identifiers | Additional information |
|---|---|---|---|---|
| Gene (Apolipoprotein E) | APOE | PMCID: PMC6106945 | HGNC:HGNC:613 | |
| Sequence-based reagent | APOE PCR Primers | Integrated DNA Technologies, Inc (IDT) | F 5'-ACA-GAA-TTG-GCC-CCG-GCC-TGG-TAC-3', R 5'-TAA-GCT-TGG-CAC-GGC-TGT-CCA-AGG-A-3' | 0.5 µL of each 50 µM Primer |
| Chemical compound, drug | FailSafe PCR Enzyme and 2X PreMix Buffers | Lucigen | FSP995J | |
| Chemical compound, drug | HHA1 | NEB | R0139S | 0.5 µL/20 µL of PCR product |
| Other | Whatman 903 Protein Saver Card | VWR | 05-715-121 | |
| Chemical compound, drug | Ultrapure Agarose | Thermo | 16500500 | 4% |
| Chemical compound, drug | GelStar Gel Stain | Lonza | 50535 | 7 µL/300 mL of gel mix |
| Chemical compound, drug | Ultra-Low Range DNA Ladder | Invitrogen | 10597012 | |
| Chemical compound, drug | Amplitaq Gold Fast Master Mix | Thermo | 4390941 | |
| Chemical compound, drug | Oragene Saliva Kit | DNAGenotek | OGR-500 | |
| Chemical compound, drug | Tris-Acetate-EDTA (TAE) 50 X (20L) | Fisher | BP1332-20 | |
| Software, algorithm | R | The R Foundation | Version 3.5.1 RRID:SCR_001905 | |
| Software, algorithm | R package, ggplot2 | Comprehensive R Archive Network (CRAN) | Version 3.1.1 RRID:SCR_014601 | |
| Software, algorithm | R package, emmeans | Comprehensive R Archive Network (CRAN) | Version 1.3.0 | |
| Software, algorithm | R package, Zelig | Comprehensive R Archive Network (CRAN) | Version 5.1.6 | |
| Software, algorithm | R package, MatchIt | Comprehensive R Archive Network (CRAN) | Version 3.0.2 | |

## Study participants

In January 2013, Phase I began with the launch of our internet-based study site at www.mindcrowd.org. Website visitors, who were 18 years or older, were asked to consent to our study before any data collection via an electronic consent form. As of 8-15-2018, we have had 256,674 non-duplicate or unique visitors to the website. Of these unique visitors, over 139,740 consented to participate. The final data set contained 59,571 participants who completed the paired associates learning (PAL) task and answered 22 demographic, lifestyle, and health questions (see *Supplementary file 2*). In addition, 973 participants, who completed Phase I and indicated they had a first-degree relative with Alzheimer's disease (FH), consented to provide a self-collected biospecimen (either dried blood or saliva) via the mail. Over 742 participants returned the biospecimen collection kits. Approval for this study was obtained from the Western Institutional Review Board (WIRB study number 1129241).

## Phase I: PAL test and demographic, medical, health, and lifestyle questions

After consenting to the study and answering an initial five demographic questions (age, sex, years of education, primary language, and country), participants were asked to complete a web-based paired-associates learning (PAL) task. For this cognitive task, during the learning phase, participants were presented 12 word-pairs, one word-pair at a time (2 s/word-pair). During the recall phase, participants were presented with the first word of each pair and were asked to use their keyboard to type (i.e., recall) in the missing word. This learning-recall procedure was repeated for two additional trials. Prior to beginning the task, each participant received one practice trial consisting of three word-pairs not contained in the 12 used during the test. Word-pairs were presented in different random orders during each learning and each recall phase. The same word pairs and orders of presentation were used for all participants. The dependent variable/criterion was the total number of correct word pairs entered across the three trials (i.e., $12 \times 3 = 36$, a perfect score).

Upon completing the PAL task, participants were directed to a webpage asking them to fill out an additional 17 demographic and health/disease risk factor questions. These questions included: marital status, handedness, race, ethnicity, number of daily prescription medications, a first-degree family history of dementia, and yes/no responses to the following: seizures, dizzy spells, loss of consciousness (more than 10 min), high blood pressure, smoking, diabetes, heart disease, cancer, stroke, alcohol/drug abuse, brain disease and/or memory problems). Next, participants were shown their results and provided with different comparisons to other test takers based on the average scores across all participants, as well as across sex, age, education, etc. On this same page of the site, the participants were also provided with the option to be recontacted for future research or not.

## Phase II: Biospecimen collection

Biospecimen collection: 4961 participants who completed phase I and had a FH of AD were solicited via email to self-collect and ship back either dried blood spots (DBS) on a 903 Protein Saver Card (Whatman, Little Chalfont, United Kingdom) or a saliva sample using an Oragene Discover kit (DNA Genotek Inc, Ottawa, Canada). Consenting participants (*N* = 973, 19.7%) were sent kits containing easy to use instructions and everything they needed to collect, and ship DBS or stabilized saliva back to our laboratory. Received DBS were stored at 4°C. Participants who unable to collect enough blood for adequate DBS were sent the saliva kit instead. For saliva, DNA was extracted and purified as per the Oragene Discover kit's instructions. Extracted DNA was stored at −20°C.

Apolipoprotein E (APOE) genotyping: DBS or extracted DNA was assayed via polymerase chain reaction (PCR, F 5'-ACA-GAA-TTG-GCC-CCG-GCC-TGG-TAC-3', R 5'-TAA-GCT-TGG-CAC-GGC-TGT-CCA-AGG-A-3') and restriction fragment length polymorphism (RFLP). For DBS, PCR was performed directly from a 1 mm punch of a DBS and the FailSafe PCR System using PreMix J and the above primers (Epicentre, Madison, WI). For extracted DNA from saliva, approximately 100 ng of DNA was added to AmpliTaq Gold Fast PCR Master Mix (Applied Biosystems: Thermo Fisher Scientific, Waltham, MA) with the above primers. After PCR, for RFLP, each sample was incubated with 0.5 mL of the HhaI (New England BioLabs, Ipswich, MA) restriction enzyme for 16 hr at 37°C. RFLP samples were run for approximately ~4 hr at 100V on a 4% agarose gel (Invitrogen, Carlsbad, CA), using GelStar Nucleic Acid Gel Stain (Lonza, #50535) and Ultra Low Range DNA Ladder (Invitrogen).

Images of the gel were acquired using a Canon G15 digital (Canon, Tokyo, Japan) camera using a DarkReader transilluminator (Clare Chemical Research Inc, Dolores, CO). APOE genotypes were then called according to previously published methods (*Oh et al., 1997*). A total of 743 FH +participants were APOE genotyped. After filtering a total of 673 were used for the final analyses. The variables were the presence or absence of an APOE ε4 allele (Carriers vs. Non-Carriers) and the Total Number of APOE ε4 alleles (i.e., 0, 1, or 2).

## Data quality control

The final dataset was filtered prior to analysis to remove participants: a) with duplicate email addresses (only 1 st entry kept), b) who did not complete all three rounds of the PAL test, c) whose primary language was not English, d) who was not between 18–85 years old, e) who did not report FH status, f) who reported a history of brain disease or memory problems, and g) whose Educational Attainment in years was not less than or equal to participants self-reported chronological age minus four.

## Statistical methods

Statistical analysis was conducted using R (version 3.5.1). For all analyses, the general linear model (GLM)/multiple regression analysis was used to model the Total Word Pairs Correct (criterion/dependent variable) as a function of our Demographic, Health, and Lifestyle Questions as well as APOE status (predictor/independent variables). Regression fit and GLM assumptions were evaluated by plotting residuals, testing: if these data were normally distributed, the presence of outliers and highly influential data points, the autocorrelation of the residuals, and the presence of multicollinearity (*Figure 1—figure supplement 1*). All measurements were taken from distinct samples. The demographic variables used for analysis were: Age, Biological Sex, Race, Ethnicity, Educational Attainment, Marital Status, Handedness, Number of Daily Prescription Medications, Seizures, Dizziness, Loss of Consciousness, Hypertension, Smoking, Heart Disease, Stroke, Alcohol/Drug Abuse, Diabetes, Cancer, and FH. For APOE analyses, the variables were the Number of APOE ε4 alleles and APOE Genotype. Due to low APOE sample size ($n$ = 519) in relation to the overall study ($N$ = 59,571), two GLMs (one per APOE variable), including only FH +participants, were evaluated. It is important to note that each of the above demographic variables and indicated interaction terms were included in every analysis unless stated otherwise due to model limitations. All plots were created with the R package, ggplot2. Since at age 65, the FH +and FH- performance converged, primary FH modifier analyses were conducted in participants $\leq$ 65. Categorical by categorical interactions were estimated using the R package, emmeans (version 1.3.0) R package. Adjustments for multiple comparisons were evaluated using Tukey's method via the emmeans package. Preprocessing with propensity score matching (PSM) was used before statistical modeling to reduce bias and variance (*Ho et al., 2007*; *Ho et al., 2011*). Matching was performed using the R package, MatchIt (version 3.0.2). Effect sizes were estimated from the matched cohort using the R package, Zelig (version 5.1.6). Zelig uses least squares regression on matched data to estimate the partial effect on an outcome of interest, in our case, total word pairs correct (*Imai et al., 2008*; *Imai et al., 2009*).

## Availability of the materials and resources

All reagents, programs/software, and other materials described herein are publicly or commercially available. A description of procedures necessary to conduct an independent replication of this research is available upon request from the corresponding author.

## Acknowledgements

### Funding

The authors wish to acknowledge the Mueller Family Charitable Trust for funding the initial creation of the MindCrowd site and to Mike Mueller for his significant donation of personal time during the development, testing, and ongoing evolution of the study. This work was also supported in part by the State of Arizona DHS in support of the Arizona Alzheimer's Consortium (PI; Eric Reiman), the Flinn Foundation (PI; Matt Huentelman), The McKnight Brain Research Foundation, The Fulbright Program, and NIH-NIA grants R01-AG041232 (Multiple-PI; to Matt Huentelman and Amanda Myers)

and R01-AG049465-05 (PI; Carol Barnes). We also acknowledge the following individual donors: Kristina Adams, Beverly Aldous, Peter Ax, Betty Barnes, Catherine Bartlett, Murali and Jori Bathina, Cathleen Becskehazy, Matt and Mary Birk, David Blake, Irwin and Marie Bliss, Sandra Bolduc, Michael and Anna Brennan, William and Tess Burleson, Christopher and Jan Cacheris, Thomas Clancy, Margaret Clark, Larry and Carol Clemmensen, Ronald Cocciole, James Conley, Linda Bowles Da Silva, Vernon Daniel, Birdeena Dapples, Virginia Davis, Brent Donaldson, Bennett and Jacquie Dorrance, Robert and Annette Dunlap, Julia Eggleston, Pat Eisenberg, Rhian Ellis, Mark Flint, Jayne Gasperson, Don and Kim Gray, Jane Hamblin, Elaine Harris, Steve and Suzi Hilton, Carolyn Horton, Jeffrey Hunt, Intel Foundation, Bill Johnsen, Marie Johnson, A.J. Kearney, Jeri Kelley, Maria Cesaria Lancaster, Bill and Julie Lavidge, Debra Lesperance, Donald Lindsay, Nate Lowrie, Marsha Luey, Terry and Gabriela McManus, Burt Moorhouse, Kathleen Parker, Brian Patrick, Nils Pearson, Lawrence Reba, Sharii Rey, Mary Riter, Lucy Roth, Victoria Ruppert, Roger Salomon, Patty Sandler, Seymour Sargent, Michael and Sheryl Sculley, Donald J Stanforth, Dean Taylor, Thomas Thimons, Paul Timm, Rex Travis, Karl Volk, Patricia Wargo, Philip Welp, Michael Wintory, Stanley Wolf, and thirty-two additional anonymous donors. Fundraising efforts for MindCrowd were led by the TGen Foundation staff, and we would like to acknowledge the efforts of Michael Bassoff and T.J. Isaacs.

### Design, testing, and administrative support

The MindCrowd site was built by The Lavidge Company (Scottsdale, Arizona) and we acknowledge their support through the donation of significant *pro bono* employee hours and resources, in particular, those provided by Stephen Heitz, Meghan Mast, and Bill Lavidge. John Tabar provided project management and volunteer hour contributions during the design and testing phases of MindCrowd. Members or past members of the Huentelman laboratory are also recognized for their conceptual contributions to MindCrowd including Chris Balak, Ashley Siniard, Adrienne Henderson-Smith, Ryan Richholt, Mari Turk, Ryan Bruhns, Claire Cambron, Amanda Wolfe, and Jim Peden. We also acknowledge the administrative support of Stephanie Althoff, Kara Karaniuk, Mary Ellen Ahearn, and Stephanie Buchholtz (for CRC and IRB activities); Galen Perry, Jeffrey Watkins and Steve Yozwiak (for marketing and communications activities); and Valerie Jones, Brian Anderson, Kati Koktavy, and Russ Brandt (for administrative coordination and project management). William Burleson, James Lowey, and Planet Argon (Gary Blessington and Robby Russell) have contributed to the current version and administration of the MindCrowd website. We acknowledge LeaseHawk (Scottsdale, Arizona) for participating in the alpha testing of MindCrowd and for providing quiet 'bunker' facilities during the drafting of this manuscript.

### Recruitment and social media

Heather Hanson, Anna Gunderson, Bryeson Rodgers, Emily Schilling, Tayllor Lillestol, and Emma Totten (and her company BuzzlyMedia) participated in social media campaigns for MindCrowd. Jennifer Jenkin and Julie Euber perform public outreach duties for MindCrowd. The Alzheimer's Association TrialMatch service for directing participants to MindCrowd and we acknowledge the support of the Alzheimer's Association Desert Southwest Chapter (Phoenix, Arizona) and the past Executive Director Deborah Schaus and current Executive Director Dan Lawler in facilitating those efforts. The Alzheimer's Prevention Initiative at Banner Alzheimer's Institute (Phoenix, Arizona) also helped with recruitment, and we acknowledge their support as well, including the Program Director Jessica Langbaum. Lastly, we would like to thank our MindCrowd Facebook friends, Twitter followers, Redditors, and other social media participants especially Lynda Carter, Ashton Kutcher, and Valerie Bertinelli for their support in the social marketing of MindCrowd.

### In memoriam

This manuscript is dedicated to the memory of our dear friend and colleague Jason Corneveaux, who originated the MindCrowd name. His contributions to our laboratory were innumerable and significant, and he continues to be sorely missed by our research team.

## Additional information

### Funding

| Funder | Grant reference number | Author |
| --- | --- | --- |
| Mueller Family Charitable Trust | | Matthew J Huentelman |

| Arizona Department of Health Services | Arizona DHS in support of the Arizona Alzheimer's Consortium | Matthew J Huentelman |
| --- | --- | --- |
| Flinn Foundation | | Matthew J Huentelman |
| National Institutes of Health | R01- AG041232 | Amanda J Myers |
| National Institutes of Health | R01-AG049465-05 | Carol A Barnes |

The funders had no role in study design, data collection and interpretation, or the decision to submit the work for publication.

### Author contributions
Joshua S Talboom, Isabelle Schrauwen, Conceptualization, Data curation, Formal analysis, Supervision, Investigation, Visualization, Methodology, Writing—original draft, Project administration, Writing—review and editing; Asta Håberg, Conceptualization, Data curation, Formal analysis, Supervision, Visualization, Methodology, Writing—original draft, Project administration, Writing—review and editing; Matthew D De Both, Conceptualization, Resources, Data curation, Software, Formal analysis, Supervision, Validation, Investigation, Visualization, Methodology, Writing—original draft, Project administration, Writing—review and editing; Marcus A Naymik, Data curation, Software, Formal analysis, Validation, Investigation, Visualization, Methodology, Writing—original draft, Writing—review and editing; Candace R Lewis, Conceptualization, Formal analysis, Investigation, Visualization, Methodology, Writing—original draft, Writing—review and editing; Stacy F Bertinelli, Conceptualization, Data curation, Project administration, Writing—review and editing; Callie Hammersland, Supervision, Project administration; Mason A Fritz, Data curation, Formal analysis, Investigation; Amanda J Myers, Conceptualization, Methodology, Writing—review and editing; Meredith Hay, Elizabeth Glisky, Lee Ryan, Conceptualization, Methodology, Writing—original draft, Writing—review and editing; Carol A Barnes, Conceptualization, Visualization, Methodology, Writing—review and editing; Matthew J Huentelman, Conceptualization, Resources, Data curation, Software, Formal analysis, Supervision, Funding acquisition, Validation, Investigation, Visualization, Methodology, Writing—original draft, Project administration, Writing—review and editing

### Author ORCIDs
Joshua S Talboom (iD) https://orcid.org/0000-0002-4327-4103
Matthew J Huentelman (iD) https://orcid.org/0000-0001-7390-9918

### Ethics
Human subjects: For all participants, informed consent, and consent to publish was obtained before study participation. This protocol and consent were approved by the Western Institutional Review Board (WIRB, protocol #20111988).

### Decision letter and Author response
Decision letter https://doi.org/10.7554/eLife.46179.017
Author response https://doi.org/10.7554/eLife.46179.018

## Additional files

### Supplementary files
• Source code 1. R-based statistical analysis and plotting script.
DOI: https://doi.org/10.7554/eLife.46179.010

• Supplementary file 1. Sample sizes. Table displaying the sample size (*n*) for each evaluated demographic, health, and lifestyle factor.
DOI: https://doi.org/10.7554/eLife.46179.011

• Supplementary file 2. Questions each participant was asked and how it was coded in R. Table displaying each question that a participant was asked on the MindCrowd website, whether the question

was asked before or after the PAL test, the question number, and how each question was coded for analysis via R (version 3.5.1).
DOI: https://doi.org/10.7554/eLife.46179.012

• Transparent reporting form
DOI: https://doi.org/10.7554/eLife.46179.013

## Data availability

The data that support the findings of this study are freely available at Dryad (https://datadryad.org) doi:10.5061/dryad.2867k2m).

The following dataset was generated:

| Author(s) | Year | Dataset title | Dataset URL | Database and Identifier |
|---|---|---|---|---|
| Talboom J, Håberg A, De Both M, Naymik M, Schrauwen I, Lewis C, Bertinelli S, Hammersland C, Fritz M, Myers A, Hay M, Barnes C, Glisky E, Ryan L, Huentelman M | 2019 | Data from: Family history of Alzheimer's disease alters cognition and is modified by medical and genetic factors | https://dx.doi.org/10.5061/dryad.2867k2m | Dryad Digital Repository, 10.5061/dryad.j1fd7 |

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
