## [Decision Letter]

Thank you for submitting your article "Family history of Alzheimer's disease alters cognition and is modified by medical and genetic factors" for consideration by *eLife*. Your article has been reviewed by two reviewers, one of whom is a member of our Board of Reviewing Editors, and the evaluation has been overseen by a Senior Editor. The reviewers have opted to remain anonymous.

As standard practice in *eLife*, the reviewers have discussed their critiques with one another in an online panel moderated by the Reviewing Editor. After consensus was reached, the Reviewing Editor has drafted this decision to help you prepare a revised submission.

Summary:

This is an interesting and potentially important study investigating the effects of family history (FH) of Alzheimer's disease on memory function across the lifespan. Using an online assessment platform (MindCrowd) the authors probed paired-associate verbal learning as an index of MTL-dependent learning from 18-85 years. The sample size of ~60,000 people is impressive, and permits a well-powered exploration of factors that potentially mediate the effects of FH on memory across the lifespan. The inclusion of APOE genotype data in a large subgroup is a further strength of the study. The most significant finding from this study is that FH of Alzheimer's disease was associated with poorer verbal paired-associate learning performance in individuals <65 years, and this effect appeared to be particularly pronounced in men. In general, the paper is well written and the findings are potentially of clinical importance, suggesting that distinct alterations on the PAL may be a useful marker of memory decline from middle-age.

Essential revisions:

1) A significant question concerns how the construct of FH was defined for participants. The exact wording for this measure must be provided as there are a number of possible options. Further, was FH related to all members in the family or not? Was FH defined in terms of both sporadic and familial AD? FH could be defined as a binary value (as in the study) or as numeric variable related to familial AD and these are important distinctions to state upfront. Please clarify throughout the manuscript.

2) The findings presented hinge upon the reliability of the self-reported data regarding FH and lifestyle conditions, however the validity and reliability of these data are not presented. Exactly what type of information was requested from participants to determine a FH of AD – important variables to consider include age of onset, specific symptom profiles, whether there was functional decline (i.e., differentiating between aMCI and AD).

Further, the reports on lifestyle characteristics are unclear. We would like to see further details regarding how the questions were phrased and how they were coded in the study. What is the reliability of these questions – is it possible that a specific answer might be interpreted and responded to differently across cultures/languages etc.?

3) While the demographic questionnaires cover a number of key factors, socioeconomic status (SES) was not explored. This is not a trivial omission as a number of the factors of interest could potentially be explained by lower SES. For example, educational attainment, dietary/medical risk factors, alcohol/drug consumption, diabetes, and even family history of AD may all relate to, or interact with, lower SES. How lower SES potentially contributes to, or interacts with, these risk factors is an important issue to address as it might be a common underlying variable that modulates the risk for cognitive decline. Further, SES may relate to selection bias of individuals who volunteer to take part in such studies. Please clarify and include these data, if possible, as a covariate of interest.

4) The potential role of other risk factors/disease does not appear to have been taken into account. Please clarify or state as a limitation.

5) The result that PAL performance was linearly associated with participant's age (Figure 2) seems not to be in agreement with findings of memory decline across age which is not described by linear functions but a sinuous curve (cf. Jack et al. on dynamics in AD development). Can the authors please comment on this discrepancy?

6) The discussion of sex-differences around the 5th decade in relation to menopause and modulating effects of estrogen is particularly interesting. Estrogen has been suggested to offer a neuroprotective effect and it would be interesting to know whether sharp declines in PAL performance are evident in women from the 6th decade onwards (when presumably hormonal levels are much lower, and hysterectomy may have occurred). This relates to the somewhat arbitrary age cut-off point of 65 years – in truncating the age-range of participants, the authors may conceal important hormonally-driven transitions in cognitive function in women. Is it possible to look at these factors earlier in the lifespan using these data?

---

## [Author Response]

Essential revisions:1) A significant question concerns how the construct of FH was defined for participants. The exact wording for this measure must be provided, as there are a number of possible options. Further, was FH related to all members in the family or not? Was FH defined in terms of both sporadic and familial AD? FH could be defined as a binary value (as in the study) or as numeric variable related to familial AD and these are important distinctions to state upfront. Please clarify throughout the manuscript.

We thank the review team for their thoughtful comments regarding FH. We recognize the importance of properly defining the FH construct for the study participant and for this reason we directly indicate what is meant by “first-degree relative” in the wording of the question on the MindCrowd site. A screenshot is included in Author response image 1 showing exactly what each test taker sees when answering this question.

We spent time reviewing prior studies on FH to come to what we consider the most accurate question to probe the potential influence of FH on cognitive performance in our cohort. Prior studies have demonstrated a stronger influence on dementia risk if the FH of Alzheimer’s disease (AD) was within a first-degree relative versus other relatives (Cannon-Albright et al., 2019). Due to this knowledge and our interest in balancing our ability to recruit a large-sized cohort with the parallel goal of collecting as much demographic, medical, and lifestyle data as would be palatable to a participant, we restricted the question regarding FH on the main MindCrowd site to that of first-degree relatives only. We added in exactly how the FH question was asked:

“FH status was determined by self-report of having (FH^+^), or not having (FH-) based on the question, “Have you, a sibling, or one of your parents been diagnosed with Alzheimer's disease?”

Like the reviewers, we also have an interest in further understanding FH risk in early vs. late AD diagnosed relatives as well as second-degree and third-degree relatives. We chose not to ask questions related to these on the main MindCrowd site to facilitate overall cohort recruitment.

However, in January of 2018, we initiated a follow-up survey questionnaire for all study participants who voluntarily provided us with contact information. In this survey, we ask additional questions related to FH – including extended family member AD diagnoses and the age of diagnosis for their first-degree relative. To date, over 6,000 individuals have answered these survey questions and we have identified 1.3% (n=84) in this survey cohort with a self-reported first-degree relative diagnosed with early-onset AD (defined for the survey participant as a diagnosis under the age of 55). We explored these data and found that early onset FH resulted in a lowered PAL performance as well – mirroring what we report in the manuscript for the first-degree FH regardless of the age of diagnosis. While the proportion of early-onset AD approximates the expected population frequency for this rare form of AD, we believe this cohort is still undersized for appropriately powered statistical analyses and therefore is beyond the scope of the current manuscript.

We have included a Supplementary file in the manuscript that lists all the demographic, medical, and lifestyle questions as they were worded for the participant as well as how the answers were coded for our statistical analyses (Supplementary file 2). In the manuscript, we include an additional Discussion of FH AD risk:

“It is important to acknowledge some limitations of our work. FH risk is known to vary depending on the relationship of the diagnosed relative, and previous reports have demonstrated that first-degree FH results in higher risk for dementia compared to second- and third-degree FH (Cannon-Albright et al., 2019). In our study, we asked about the first-degree FH only; therefore, it is possible that individuals who have other FH risk from extended family members were included in our non-FH group. Additionally, it is possible that the form of AD, late-onset versus the rarer early-onset form, may encode different levels of FH risk. Future work is planned to investigate the FH effect in the study cohort, including an improved ability to separate late and early onset FH for each participant as well as to inquire about additional extended family member FH status.”

2) The findings presented hinge upon the reliability of the self-reported data regarding FH and lifestyle conditions, however the validity and reliability of these data are not presented. Exactly what type of information was requested from participants to determine a FH of AD – important variables to consider include age of onset, specific symptom profiles, whether there was functional decline (i.e., differentiating between aMCI and AD).Further, the reports on lifestyle characteristics are unclear. We would like to see further details regarding how the questions were phrased and how they were coded in the study. What is the reliability of these questions – is it possible that a specific answer might be interpreted and responded to differently across cultures/languages etc?

We agree that self-report data are reliant on the accuracy of each of those self-reported variables. We did not specifically study the reliability of our particular self-report data as there have been many studies to examine this question previously (Maukonen, Mannisto, and Tolonen, 2018; Nikolaou, Hankey, and Lean, 2017). The conclusion from these previously published examinations of self-report reliability is that the error of the self-reported answer is unique for each variable and can be related to the complexity of the question and answer options (e.g. dichotomous vs. quantitative) as well as to social influences related to the answer, however, the range of error reported in these studies was between 0.3-20% for quantitative traits like height and weight (Elgar and Stewart, 2008; Ikeda, 2016; Shields et al., 2008; Yoon et al., 2014). We do believe it is possible that a self-report answer could also vary across cultures, as the review team pointed out; however, we would like to emphasize that only individuals who self-declared English as their primary language were included in the analyses reported in the manuscript. Therefore, we believe that this potential cultural effect is likely minimized in the cohort used for analysis in our manuscript.

We addressed the details of the wording and rationale for our FH question above in the answer to Comment #1 and we agree with the reviewers that publishing exactly how each demographic, medical, and lifestyle question was asked and how the data were coded during statistical analysis is important and we have now included all of these details in Supplementary file 2.

We performed two additional analyses to test the impact of the self-report error on the FH finding. First, we re-analyzed the FH effect after introducing additional error into the coding of the FH self-report response. This was done by randomly re-assigning FH status to various percentages of the MindCrowd cohort (stepwise from 2-30% of individuals) and re-analyzing the effect of FH using our complete statistical model. This was performed a total of 10,000 times for each error percentage and the results on the p-value were reported using boxplots. It is important to note that this error is an additional amount introduced on top of whatever may already be present in the MindCrowd data due to self-report of FH. With 8% additional introduced error we are able to show that we would still report a statistically significant effect of FH on PAL in 100% of the 10,000 tested iterations. An additional 24% error in FH status would still result in a statistically significant effect of FH on PAL in over 50% of the 10,000 iterations. These data suggest that it is highly unlikely that FH self-report error is driving the significant effect of FH on PAL we report in the manuscript.

Secondly, we examined the FH effect on PAL through the use of permutation testing. This is an approach utilized to determine the probability of a false positive finding under the null hypothesis. The FH data label for every participant was randomly assigned and the t-statistic for the FH effect on PAL was re-calculated. This was performed one million times. In every case, not a single t-statistic arising from a permutation of the FH data label was observed to be more extreme than our reported t-statistic (permuted p-value = <1e-6). This suggests that the odds of our FH finding being observed due to chance alone is less than one in one million.

These findings are illustrated in Author response image 2. Figure A from Author response image 2 was added to the manuscript as “Figure 1—figure supplement 2”.

**Author response image 2. respfig2:** Graphs exploring the influence of additional self-report error on the FH-AD effect. (**A**) Simulated additional self-report error and the impact on the significance of the FH effect in MindCrowd. The effect of error in self-reported FH-AD status was simulated by adding additional error to the MindCrowd cohort by re-assigning individual FH responses between 1-30%. and re-running the full statistical model. This was repeated 10,000 times for each error rate. Boxplots representing the distribution of p-values for the re-analysis under the new error model are illustrated. This demonstrates that even with 8% additional error added to the self-report FH question, we would still have identified a significant association with FH in all 10,000 cases. Even with 24% additional FH self-report error we would have still reported a significant association between FH and PAL over 50% of the time (better than by chance alone). Therefore, this suggests that even with significant levels of additional error in FH self-report the association between FH and PAL performance would still have been noted by our study. (**B**) Permutation testing of the FH effect on PAL.The p-value describing the FH effect on PAL in MindCrowd was permuted to determine the probability of a false positive under the null hypothesis. The vertical dashed red line indicates our reported t-statistic from the manuscript and the histogram of permuted t-statistics are indicated in black bars. After one million permutations of the FH data label, not a single t-statistic arising from a permutation of the data were observed to be more extreme than our reported t-statistic (permuted p-value = <1e-6). This suggests that the odds of our finding being observed due to chance is less than one in one million.

Further, while acknowledging the potential drawbacks with web-based self-report approaches, we agree with the authors of a recent book chapter who suggest some advantages to web-based self-report data collection compared to traditional in-person methods, "The presence of the researcher at the completion of the questionnaire may affect answers as well. For example, subjects may change their behavior or demonstrate an improvement in their outcome because they know they are being observed. More specifically, in the presence of the examiner, a responder may not feel comfortable selecting the extreme choices (Demetriou, Ozer, and Essau; 2015).” This is a known effect of self-report data collected in the presence of a medical or research professional. In short, societal pressures can lead to over- and under-reporting of certain variables (e.g. reporting less body weight) and there is suggestive evidence that the anonymity afforded by an internet self-report collection approach may actually lead to lower rates of self-report error due to this fact.

The addition of self-report error is added as a potential limitation to our study in the Discussion section of the manuscript:

“Due to the large, distributed, and electronic nature of our study cohort, we rely on self-report answers to demographic, lifestyle, and health questions. Current studies comparing self-report data given over the internet versus in-person collected data show anywhere from a 0.3-20% discrepancy for height and weight measurements (Maukonen, Mannisto, and Tolonen, 2018; Nikolaou, Hankey, and Lean, 2017). To investigate the potential role that such error may play on our FH AD effect, we re-analyzed the FH effect after introducing additional error into the coding of the FH self-report response. Additional error was added by randomly re-assigning FH status to various percentages of the cohort (stepwise from 2-30% of individuals) and re-analyzing the effect of FH using our complete statistical model. This was performed a total of 10,000 times for each error percentage, and the resulting influences on the p-value were reported using boxplots. With 8% additional introduced error we are able to show statistically significant effects of FH on PAL in 100% of the 10,000 tested iterations while an additional 24% error in FH status would still result in a statistically significant effect of FH on PAL in over 50% of the iterations (see Figure 1—figure supplement 2). These results suggest that it is unlikely that FH self-report error is driving the significant effect of FH on PAL.”

3) While the demographic questionnaires cover a number of key factors, socioeconomic status (SES) was not explored. This is not a trivial omission as a number of the factors of interest could potentially be explained by lower SES. For example, educational attainment, dietary/medical risk factors, alcohol/drug consumption, diabetes, and even family history of AD may all relate to, or interact with, lower SES. How lower SES potentially contributes to, or interacts with, these risk factors is an important issue to address as it might be a common underlying variable that modulates the risk for cognitive decline. Further, SES may relate to selection bias of individuals who volunteer to take part in such studies. Please clarify and include these data, if possible, as a covariate of interest.

We agree with the review team that SES is an essential factor to consider. Our initial 26 demography and health questions included educational attainment, but not income. Hence, it is not possible to evaluate SES across all of our participants. However, all analyses included education as well as all other covariates (see Supplementary file 2) in the statistical model.

Since the submission of this manuscript, we have been collecting responses from a follow-up survey allowing us to calculate both current (your current income and educational attainment) and childhood SES (your parent’s income and educational attainment). Initial evaluations of current and adult SES within this cohort of 6,187 individuals did not reveal a statistically significant association between the SES composite and PAL. The individual factors that contribute to the SES composite score – e.g. the income and educational attainment values – were also tested individually and only educational attainment was associated with PAL performance.

For reference, included as Author response image 3 is a figure illustrating the current income distributions within the MindCrowd survey cohort demonstrating that ~12% of the cohort has a current income equal to or less than $35,000. This amount approximates the U.S. national estimate for poverty rate (reported as 13.5% in 2015 by U.S. Census Bureau; 2017).

**Author response image 3. respfig3:** 

Further, we acknowledge that SES is a complex measure which is not defined the same across all disciplines (e.g., includes or excludes measures of education quality, access to health care, employment, health variables, etc.), and there appears to be marked cultural differences as to which factors best describe the SES construct as well (e.g., employment in the UK, see Rose et al., 2005). Because our cohort includes participants from all over the world, and the second largest portion of our data (n = 2060), next to the US, is based on participants in the UK, we think that traditional SES measurements may not adequately capture the SES construct in our data.

We now address the potential limitation of not measuring SES in our cohort in the Discussion section. Please see the response to review team critique #4 below for the exact added wording we utilize.

4) The potential role of other risk factors/disease does not appear to have been taken into account. Please clarify or state as a limitation.

We want to note for the review team that when a variable was tested for association with PAL, we included other collected demographic, lifestyle, and medical factors as covariates in order to control for their influence. In re-reading the description of the statistical methods it is evident that this fact wasn’t fully explained. To address this, we have updated the Materials and methods section:

“It is important to note that each of the above demographic variables and indicated interaction terms were included in every analysis unless stated otherwise due to model limitations.”

However, we do agree that it is possible that factors that may influence PAL were not collected via our list of demographic, lifestyle, and medical factors. We have added this limitation in the Discussion section:

“It is likely that we did not measure all demographic, lifestyle, and health factors that are associated with differential PAL performance. One such example is socioeconomic status (SES). SES has been shown to have an association with brain structure and cognitive measures during development (reviewed in Brito and Noble, 2014) and work also suggests SES could play a role in AD risk (Qian, Schweizer, and Fischer, 2014; Stepkowski, Wozniak, and Studnicki, 2015) reviewed in Seifan and colleagues (2015). Importantly, while we did not measure SES directly, we did assess factors commonly used to construct the SES composite (e.g., Educational Attainment). In addition, due to the international recruitment of our study cohort, normalization of the SES construct is complicated due to differing definitions of the factors used to calculate SES across nations (Rose, Pevalin, and O'Reilly, 2005).”

5) The result that PAL performance was linearly associated with participant's age (Figure 2) seems not to be in agreement with findings of memory decline across age which is not described by linear functions but a sinuous curve (cf. Jack et al. on dynamics in AD development). Can the authors please comment on this discrepancy?

The direct comparison with Jack et al. is problematic because those data (and the resulting model) are based on disease progression. Their model indicates how AD-relevant biomarkers can increase to non-normative levels decades before the cognitive decline is clinically registered. Therefore, the alteration of these disease-associated biomarkers could alter the cognitive decline curve for a disease cohort but may not be present – or if they are present, may exert a different magnitude of effect in a normative aging cohort like MindCrowd. We agree with the review panel that the linear decline with age in MindCrowd was surprising; however, it is clearly the best fit for what we observe in the study. In fact, another cross-sectional study of 1250 participants noted a similar linear decline in PAL performance (i.e., from the Wechsler Memory Scale III) from 18 to 80 years of age (Salthouse, 2003).

Future work leveraging longitudinal assessments of large normative aging cohorts like MindCrowd will help to further elaborate on the aging trajectory and will also help us better appreciate individual deviations from this trajectory. We view the non-longitudinal nature of MindCrowd as a current limitation. Indeed, a longitudinal study of our cohort may reveal a non-linear age-related decline in PAL performance. We have discussed this limitation further in the manuscript:

“Lastly, PAL was tested cross-sectionally in the cohort; therefore, determinations about the influence of collected factors on trajectories of change in performance across time within an individual subject are not possible. Additional longitudinal-based studies will be necessary to identify this class of variables.”

6) The discussion of sex-differences around the 5th decade in relation to menopause and modulating effects of estrogen is particularly interesting. Estrogen has been suggested to offer a neuroprotective effect and it would be interesting to know whether sharp declines in PAL performance are evident in women from the 6th decade onwards (when presumably hormonal levels are much lower, and hysterectomy may have occurred). This relates to the somewhat arbitrary age cut-off point of 65 years – in truncating the age-range of participants, the authors may conceal important hormonally driven transitions in cognitive function in women. Is it possible to look at these factors earlier in the lifespan using this data?

We are in agreement with the review team that the sex-difference around the 5th decade of life is exciting; however, we are confused by the review team comment regarding the cut-off point of 65 years for this sex effect. In Figure 2 in the manuscript, we demonstrate the sex effect across all ages 18 and above in the study.

The review team is likely referring to the age cut-off we utilized for the FH analysis. This was rationalized in the manuscript (see quote below) due to the convergence of the trend lines for the FH effect. We did not employ an age cut-off for any of the other analyzed variables, including sex.

“Indeed, the linear trend lines for diabetes cross at 50 years of age (data not shown) and at age 65 for FH (Figure 4A.). Due to the significant Age x FH interaction, our analyses evaluating interactions with FH included only participants ≤65 years old.”

In our follow-up survey mentioned earlier, we have started to look at menopause (hormone therapy and surgery options) and hormonal contraception choices. Due to the lower size of the female survey cohort (~4,000 female respondents to date), we think that the results of the survey thus far are outside of the realm of this particular publication. However, a further discussion of this effect and how it will be addressed in future studies was expanded in the Discussion section:

“It is interesting that our study found the associated disparity between women’s and men’s PAL scores enlarged around the 5th decade of life. The 5th decade of life is the approximate age when women undergo menopause in developed countries. Menopause-related changes to women’s hormonal milieu, either endogenously or via hormone treatment or gynecological surgery have been found to alter cognition during this period (discussed in Koebele and Bimonte-Nelson, 2016). Future studies of this cohort will dissect which medical choices at menopause, and medical choices earlier in a woman’s lifespan, may underlie better preservation of verbal memory in middle aged women as compared to men.”

References

Demetriou, C., Ozer, B. U., and Essau, C. A. (2015). Self-Report Questionnaires. In The Encyclopedia of Clinical Psychology.

Elgar, F. J., and Stewart, J. M. (2008). Validity of self-report screening for overweight and obesity. Evidence from the Canadian Community Health Survey. Can J Public Health, 99(5), 423-427.

Ikeda, N. (2016). Validity of Self-Reports of Height and Weight among the General Adult Population in Japan: Findings from National Household Surveys, 1986. PloS one, 11(2), e0148297. doi:10.1371/journal.pone.0148297

Jack, C. R., Jr., Knopman, D. S., Jagust, W. J., Shaw, L. M., Aisen, P. S., Weiner, M. W.,. . . Trojanowski, J. Q. (2010). Hypothetical model of dynamic biomarkers of the Alzheimer's pathological cascade. Lancet Neurol, 9(1), 119-128. doi:10.1016/S1474-4422(09)70299-6

Jack, C. R., Jr., Knopman, D. S., Jagust, W. J., Petersen, R. C., Weiner, M. W., Aisen, P. S.,. . . Trojanowski, J. Q. (2013). Tracking pathophysiological processes in Alzheimer's disease: an updated hypothetical model of dynamic biomarkers. Lancet Neurol, 12(2), 207-216. doi:10.1016/S1474-4422(12)70291-0

Salthouse, T. A. (2003). Memory aging from 18 to 80. Alzheimer Dis Assoc Disord, 17(3), 162-167.

Shields, M., Connor Gorber, S., and Tremblay, M. S. (2008). Estimates of obesity based on self-report versus direct measures. Health Rep, 19(2), 61-76.

U.S. Census Bureau data (2017); U.S. Department of Commerce, Bureau of Economic Analysis, Survey of Current Business; and DataQuick Information Systems, a public records database company located in La Jolla, San Diego, CA.

Yoon, K., Jang, S. N., Chun, H., and Cho, S. I. (2014). Self-reported anthropometric information cannot vouch for the accurate assessment of obesity prevalence in populations of middle-aged and older Korean individuals. Arch Gerontol Geriatr, 59(3), 584-592. doi:10.1016/j.archger.2014.08.008